# An Updated Narrative Mini-Review on the Microbiota Changes in Antenatal and Post-Partum Depression

**DOI:** 10.3390/diagnostics12071576

**Published:** 2022-06-28

**Authors:** Bogdan Doroftei, Ovidiu-Dumitru Ilie, Roxana Diaconu, Delia Hutanu, Irina Stoian, Ciprian Ilea

**Affiliations:** 1Faculty of Medicine, University of Medicine and Pharmacy “Grigore T. Popa”, University Street, No. 16, 700115 Iasi, Romania; bogdandoroftei@gmail.com (B.D.); stoian.irinalv@yahoo.com (I.S.); cilea1979@yahoo.com (C.I.); 2Clinical Hospital of Obstetrics and Gynecology “Cuza Voda”, Cuza Voda Street, No. 34, 700038 Iasi, Romania; dr.diaconuroxana@gmail.com; 3Origyn Fertility Center, Palace Street, No. 3C, 700032 Iasi, Romania; 4Department of Biology, Faculty of Biology, “Alexandru Ioan Cuza” University, Carol I Avenue, No. 20A, 700505 Iasi, Romania; 5Department of Biology, Faculty of Chemistry-Biology-Geography, West University of Timisoara, Vasile Pârvan Avenue, No. 4, 300115 Timisoara, Romania; delia.hutanu@e-uvt.ro

**Keywords:** antenatal depression, post-partum depression, microflora, gut-brain axis, behavior

## Abstract

Background: Antenatal depression (AND) and post-partum depression (PPD) are long-term debilitating psychiatric disorders that significantly influence the composition of the gut flora of mothers and infants that starts from the intrauterine life. Not only does bacterial ratio shift impact the immune system, but it also increases the risk of potentially life-threatening disorders. Material and Methods: Therefore, we conducted a narrative mini-review aiming to gather all evidence published between 2018–2022 regarding microflora changes in all three stages of pregnancy. Results: We initially identified 47 potentially eligible studies, from which only 7 strictly report translocations; 3 were conducted on rodent models and 4 on human patients. The remaining studies were divided based on their topic, precisely focused on how probiotics, breastfeeding, diet, antidepressants, exogenous stressors, and plant-derived compounds modulate in a bidirectional way upon behavior and microbiota. Almost imperatively, dysbacteriosis cause cognitive impairments, reflected by abnormal temperament and personality traits that last up until 2 years old. Thankfully, a distinct technique that involves fecal matter transfer between individuals has been perfected over the years and was successfully translated into clinical practice. It proved to be a reliable approach in diminishing functional non- and gastrointestinal deficiencies, but a clear link between depressive women’s gastrointestinal/vaginal microbiota and clinical outcomes following reproductive procedures is yet to be established. Another gut-dysbiosis-driving factor is antibiotics, known for their potential to trigger inflammation. Fortunately, the studies conducted on mice that lack microbiota offer, without a shadow of a doubt, insight. Conclusions: It can be concluded that the microbiota is a powerful organ, and its optimum functionality is crucial, likely being the missing puzzle piece in the etiopathogenesis of psychiatric disorders.

## 1. Introduction

AND, also known as prenatal or perinatal depression alongside postnatal or PPD, reunited under perinatal mood and anxiety disorders (PMADs), is an umbrella that comprises psychiatric episodes, with an overall prevalence of 17% in PPD [1] and up to 20% in AND [2]. However, the interval between the undergoing pregnancy and the first month follow-up could represent the transition to a major depressive disorder (MDD), classified as peripartum depression according to the DSM 5 [3,4].

The associated phenotype in advanced stages exert antagonistic effects on the patient’s psychological and physiological profile. Precisely, suicidal ideation is a significant co-founder of mortality among women [5]. Left untreated, it negatively impacts the cognitive and socio-emotional development of the newborn [6,7]. Unfortunately, no curative strategy involving prevention, diagnosis, and treatment proved reliable [8].

Beside these aspects stands a shorter breastfeeding duration [9], but prior to this is the intrauterine environment and subsequent risk of compromising the infant’s immune system [10]. Thus, researchers emphasize the possible role of the women’s microbiota as a biological factor in the pregnancy progress [11].

There are numerous shaping factors responsible for the loss of eubiosis. Among the plethora of them are delivery methods, antibiotics, antidepressants, and child nutrition, which are definitive [12]. Breastfeeding promotes shifts in bacterial ratio and ensures the proliferation of the so-called “beneficial” microorganisms, even in small amounts [11]. 

Considering the multifaceted and branched involvement of the microscopic entities harbored by almost every constitutive site of the human body, the present narrative mini-review aims to offer an update on the current discoveries surrounding microflora and behavioral changes in women suffering from AND PPD.

## 2. Methodology

This narrative mini-review respects the standard protocol described by Green et al. [13].

### 2.1. Database Search Strategy

The literature databases explored for information until inception (May 2022) were *PubMed/Medline*, *ISI Web of Knowledge*, *ScienceDirect*, and *Scopus*. Several keywords, including “microbiota/microflora” in combination with “antenatal depression”, “perinatal depression“, “prenatal depression”, “postnatal depression”, and “postpartum depression”, were employed during database tracking.

### 2.2. Inclusion Criteria

We included only articles that reported experience on human patients or experimental models in mice, rats, and zebrafish (*Danio rerio*).

### 2.3. Exclusion Criteria

Case report(s)/series, meta-analyses, review(s), standard or systematic, articles written in another language than English, letters to the editor, conference posters, work protocols, preprints, and computational simulations were not considered suitable.

### 2.4. Study Selection

Four independent authors (O.-D.I., R.D., D.H., I.S.) screened the titles and abstracts of the retrieved entries. We completed the assignment of all literature based on title, abstract, and full content. Any discrepancy was solved by consent with the remaining two authors (B.D., C.I.).

## 3. Results

A total of 1558 articles were returned in the interval 2018–2022. Per database searched, we identified 56 in *PubMed/Medline*, 59 in *Scopus*, 83 in *ISI Web of Knowledge*, and 1360 in *ScienceDirect*. Divided in chronological order, *n* = 17, *n* = 17, *n* = 29, and *n* = 332 were conducted concerning PPD, while *n* = 39, *n* = 42, *n* = 54, and *n* = 1028 for AND. A total of *n* = 47 articles met the eligibility criteria.

Table 1 contains a stratification that strictly analyzes the changes occurred in microflora composition. From *n* = 10 studies that sought to demonstrate a link between different stressors and subsequent development along all three stages of pregnancy, only *n* = 7 brought evidence into microflora translocations.

In order to offer a comprehensive overview, we found it suitable to mark each Section from this mini-review with “*”, “**”, or “***”, indicating either experiments conducted strictly on humans, from a combined perspective (humans + experimental models), or strictly on experimental models. To maintain continuity with a logical argument, we summarize studies marked with “**” in a table that contains the key observations made by the authors (Table 2).

### 3.1. Microbiota Shaping Factors in Non- and Pregnant Females **

Prenatal stress inhibits the growth and development of neurons [14,15,21]. Thus, behind cytokine elevation and neurotransmitters depletion might be the involvement of the tryptophan pathway [16,22,23]. There is an interconnection between bacterial shifts [17,18,19,20] and how microglia respond to environmental stressors in a time- and sex-dependent manner [24,25]. CD4T initiates transcriptional maturation in microglia, and in its absence, it triggers a defective synaptic behavior [26]. Kang et al. [27,28] discussed in two reports an irrespective link between breastfeeding and secretory immunoglobulin A (sIgA) in infants born of mothers with antenatal depression. The concentrations of anti-inflammatory cytokines IL-1β and IL-10 are higher in Caesarean (C)-section-born individuals by comparison with those born vaginally [29], while maternal precarity and maternal immune activation (MIA) has low diversity and high relative abundance of *Enterobacteriaceae* and *Streptococcaceae* next to the low relative abundance of *Bifidobacterium* and *Lachnospiraceae* [30,31].

#### 3.1.1. Probiotics **

Probiotics are the most efficient vectors used to re-establish disruption of the gut–brain axis (GBA), mitigating lipopolysaccharide (LPS) programming in a sex-dependent manner the hypothalamic–pituitary–adrenal (HPA) axis [32]. Early exposure causes a decrease in *Lactobacillus* genera, with *L. reuteri* replenishment countering bacterial endotoxins action [33]. Even though the first signs of improvements arose approximately half a day post-administration, the effect in females is limited in contrast to their counterparts [34]. *Lactobacillus paracasei* DTA 83 [35], *Lactobacillus murinus* HU-1 [36], *Lactobacillus rhamnosus* [37], and *Lactobacillus casei* [38] reverse atypical neurodevelopmental abnormalities in offspring and carriers. Maternal Bimuno^®^ galacto-oligosaccharides (B-GOS) increase short-chain fatty acids (SCFAs) levels [39], but there were situations where probiotics or other supplements such as n-3 long-chain polyunsaturated fatty acids (n-3 LC-PUFAs) and ecologic barrier had modest effects [40,41]. The symbiotic microorganisms within Tibetan kefir grains (TKGs) may offer robust gastrointestinal (GI) immunity due to enriched *Lactobacillus kefiranofaciens* and *Lactobacillus helveticus* [42]. Another reliable approach to reconstructing GI flora represents the synergic effect with psychobiotics ADR-159 [43] or with pre- (α-lactalbumin) and postbiotics (sodium butyrate) [44]. There are two registered clinical trials (NCT04741971, NCT04472065) underway. They are estimated to be completed in 2024. Hopefully, they will offer insight into diminishing psychiatric symptoms. Unfortunately, the estimated number of people enrolled is quite small (*n* = 76, *n* = 220), currently being limited to a single center (where the case is being studied).

#### 3.1.2. Breastfeeding **

The human milk incorporates a multitude of bioactive structures, especially *Lactobacillus*. Interestingly, psychosocial distress does not influence the bacterial ratio in the first trimester post-delivery but rather the milk content [45], with recent evidence suggesting changes in the milk metabolome [46]. Early limited formula feeding followed by breastfeeding reduces the risk of infant mortality and the likelihood of readmission [47]. For example, milk fat globule membrane (MFGM) modestly secures the persistence of homeostasis in rats, particularly visceral hypersensitivity [48].

#### 3.1.3. Diet ***

The ketogenic diet (KD) influences brain volume in a non-selective manner [49]. A high-fiber diet could overcome obesity-induced cognitive and metabolic changes following a hypercaloric diet [50,51]. An optimum intake upregulates neurotransmitters expression and prevents neuroinflammation. Operational taxonomic units (OTUs) that belong to *Bacteroidales*_S24-7 and *Lactobacillus* enhance neuronal plasticity and SCFAs levels [52].

**Table 2 diagnostics-12-01576-t002:** Comprehensive overview of the studies performed in the Section 3.1 based on the studied model.

Key Observations	Reference
Section 3.1.	
*Rats*	
- 5-HT decrement in serum, prefrontal cortex, and hippocampus; - Kyn/Trp ratio elevation in serum and prefrontal cortex; - Up-regulation of IDO in colon and brain; - Decreased expression of TPH; - *Firmicutes*↓, *Bacteroidetes* ↓, *Clostridium XlVa* ↑, and *Ruminococcus* ↑, both positively and negatively correlated with 5-HT level;	[16]
- Hindered expression levels of MR, BDNF, IGF1R, GR, NEUN, GFAP, CRHR1, and GABAB2 in the hippocampus and cell immunity; - Reduced ratio of CD4 and CD8 lymphocytes from the peripheral blood;	[21]
*Mice*	
- Impaired social behavior; - Increased CORT; - Neuroinflammation; - Decreased OXT; - Decreased 5-HT in the cortex; - *Bacteroides* and *Parabacteroides* ↓;	[14]
- Fewer neural progenitors and newborn neurons; - Modified gene expression in the cortex of embryonic subjects; - Abnormal responses to anxiety- and depression-like behaviors; - Microbial structure disturbance;	[15]
- Social deficits and increased anxiety; - Differential regulation of *Arc*, *Btg2*, *Fosb*, *Egr4,* or *Klf2* in a sex-dependent manner; - *Lachnospiraceae* ↓, *Porphyromonadaceae* ↓, *Mucispirillum* ↓, *Firmicutes* ↑, *Bacteroides* ↑, *Lactobacillus* ↑↓, *Alloprevotella* ↑;	[24]
- Microglia experiencing differentiation grades, palpable by transcriptomic signatures and chromatin accessibility topography; - The microglia in GF mice had a time and sexually dimorphic result; - Antibiotics initiate biased sexual microglial reactions by inducing long-term effects on microbiota; - The overlap between human fetal microglia with murine transcriptomic signature;	[25]
- Lack of CD4 T cells in murine models suspends microglia in a position between fetal and adult stage; - Defects within this component cause an excess of immature neuronal synapses, followed by behavioral disturbances;	[26]
- Social deficits but pronounced in NIH Swiss mice; - Changes in V1AR mRNA expression in the hypothalamus in NIH Swiss mice;	[31]
*Humans*	
- Firmicutes ↓, Faecalibacterium ↓, Phascolarctobacterium ↓, Butyricicoccus ↓, Lachnospiraceae ↓, Enterobacteriaceae ↓ in PDD patients; - Correlation of Phascolarctobacterium, Lachnospiraceae, Faecalibacterium, and Tyzzerella.3 with depressive symptoms severity;	[17]
- *Citrobacter**↑*in infants born to mothers previously exposed to IPV; - *Enterobacteriaceae**↑*, *Veillonellaceae* ↓ at 20–28 weeks of life and *Gammaproteobacteria* ↓ over time; - *Weissella**↑*at 4–12 weeks; - *Lactobacillaceae**↑*and *Peptostreptococcaceae**↓*in the feces at birth; - No changes in β-diversity;	[18]
- Positive association between FDR of chronic PPD mothers with *Proteobacteria* phylum in infants; - Negative association between SCL, PRAQ-R2, and daily hassles negative scale with *Akkermansia*; - Negative associations between HCC and *Lactobacillus*; - Negative association between chronic PDD, nor HCC with microbial diversity;	[19]
- α-diversity of mother’s microbiota during the first trimester predicts introspection behavior at two years of age; - Abundance of *Lachnospiraceae* and *Ruminococcaceae* in mothers of children with normative behavior; - Prenatal diet might indirectly influence internalizing behavior via higher α-diversity of mother fecal microbiota;	[20]
- Low Kyn levels associations with high depressive symptoms; - Prenatal Trp and Kyn/Trp ratio regulation potential between IL-6 and depressive symptoms; - No correlation between Trp and Kyn and cortisol on depressive symptoms;	[22]
- Changes in the Kyn metabolism is associated with baseline primary and secondary bile acids; - Positive association between secondary bile acids and UDCA and derived conjugates with low bacterial diversity, particularly of *Lachnospiraceae* crucial to SCFAs production; - Positive correlation between history of anxiety with UDCA levels but not with depression; - A low dietary fiber in patients having a history of anxiety or depression;	[23]
- Low sIgA concentration in infants born to mothers with pre- and postnatal symptoms;	[27]
- Low sIgA in infants born to mothers with antepartum and persistent depression;	[28]
- C-section born individuals exhibit high vulnerability to acute stress; - High concentration of IL-1β and IL-10 following C-section;	[29]
- Positive association between maternal precarity with bacterial communities; *Enterobacteriaceae* *↑*, *Streptococcaceae* ↑, *Bifidobacterium* ↓, and *Lachnospiraceae* ↓; - *Veillonella*↑ in case of HPA axis dysregulation;	[30]
Section 3.1.1.	
*Rats*	
- *Lactobacillus casei*-based diet improved depressive-like phenotype; - *Lactobacillus casei* rescued BDNF, NR1, ERK1/2, and monoamines;	[38]
*Mice*	
- Pubertal LPS irremediably declined GR expression in the PVN; - Pubertal probiotics block LPS-induce reduction in GR;	[32]
- Depletion of *Lactobacillus* in LPS-exposed subjects; - *Lactobacillus reuteri* replenishment regulate LPS-induced sickness and correct memory dysfunction, anxiety-like, and stress-reactivity panel;	[33]
- Diminished LPS-induced changes of body weight at 12 and 48 h; - Probiotics alleviate LPS-induced exacerbation of central cytokine mRNA expression in the following structures: hypothalamus, hippocampus, and PCF 8 h post-intervention; - Alteration of TLR-4 in PVN due to probiotics regime;	[34]
- *Lactobacillus paracasei* DTA 83 modulate the expression of GAD 65, GAD 67, and GABAA receptor α3 subunit in the hippocampus;	[35]
- *Lactobacillus murinus* HU-1 rescue behavioral disturbances and region-specific microglial activation; - Maternal microbiome dysbiosis increases the expression of Trp53 and Il1β and Cx3cr1 protein in the prefrontal cortex;	[36]
- *Lactobacillus rhamnosus* in combination with sham surgery diminished anxiety-like phenotype and re-established HPA axis connectivity; - Increase of splenic T regulatory cells and decrease in activated microglia in the hippocampus; - Vagotomy advanced anxiolytic outcomes , HPA modulation, and growth in T regulatory cells;	[37]
- B-GOS boosted exploratory behavior in parallel with a reduction in the expression of GLU receptors in the hippocampus; - B-GOS increased cortical GLU subunits and social preferences and diminished anxiety;	[39]
- ADR-159 diet improved social component of fed animals and decreased CORT levels;	[43]
- In autistic model, ALAC and NaB enhanced animal conviviality and memory in the PA evaluation; - In CUMS mice, promoted improvements in depressive-like behavior in the FST and sucrose preference test and also enhanced memory and learning in the PA, NOR, and Morris water maze tests;	[44]
*Humans*	
- Mediocre improvements in depressive symptoms following the administration of FO at 3, 6, and 12 months follow-up;	[40]
- Lack of ecologic barrier efficiency in the study group compared with placebo after 2 months investigation;	[41]
Section 3.1.2.	
*Rats*	
- MS caused visceral hypersensitivity; - MFGM ameliorated MS-related visceral hypersensitivity; - Both MS and MFGM supplementation had modest effects on spatial memory; - There was no effect of MS on enteric neuronal or glial networks, but an increased immunoreactivity of class III β-tubulin in colonic myenteric ganglia was observable in MFGM non-separated group;	[48]
*Humans*	
- *Lactobacillus* ↑ and *Staphylococcus* ↓ in the first three months postnatal; no relative abundance between women; - Fluctuations in *Firmicutes*, *Proteobacteria*, *Bacteroidetes*, *Acinetobacter*, *Flavobacterium*, and *Lactobacillus* at the phylum and genera in women with low psychological distress; - Low microbial diversity in high in contrast with low maternal psychological distress in the first three months post-delivery;	[45]
- Positive association between prenatal psychological distress and concentration of caprate and hypoxanthine; - Positive association between milk cortisol and lactate concentration;	[46]
- ELF did not impact *Lactobacillus* or *Bifidobacterium* abundance compared with control; - ELF did not amplify the risk of *Clostridium* expansion;	[47]
Section 3.1.4.	
*Humans*	
- SSRI could elevate GI dysfunctionalities in individuals in childhood;	[53]
*Rats*	
- Metabolite bioavailability accompanied by fecal microbiota diversity due to pregnancy and lactation segregation; - Low fecal amino acid concentration during the transition period from pregnancy to lactation due to fluoxetine treatment; - Negative correlation between amino acid concentration and the relative abundance of *Prevotella* and *Bacteroides*;	[54]
*Mice*	
- Fluoxetine caused widespread abnormalities in transcriptome of the fetal brain during mid-gestation; - Oscillations of genes expression involved in synaptic organization and neuronal signaling, DNA replication, and mitosis; - Antibiotics depleted gut composition and induced transcriptional responses of the fetal brain to fluoxetine treatment in females; - Prevention due to antibiotics of OPCML in the thalamus and lateral ganglionic eminence of the fetus.	[55]

5-HT, serotonin; Kyn, kynurenine; Trp, tryptophan; IDO, indoleamine 2,3-dioxygenase; TPH, tryptophan hydroxylase; MR, mineralocorticoid receptor; BDNF, brain-derived neurotrophic factor; IGF1R, insulin-like growth factor 1 receptor; GR, glucocorticoid receptor; NEUN, neuronal nuclei; GFAP, glial fibrillary acidic protein; CRHR1, corticotropin-releasing hormone receptor 1; GABAB2, gamma-aminobutyric acid B receptor 2; CORT, corticosterone; OXT, oxytocin; V1AR, vasopressin receptor 1a; IPV, intimate partner violence; FDR, false-discovery rate; SCL, Symptom Checklist-90; PRAQ-R2, Pregnancy-Related Anxiety Questionnaire—Revised 2; HCC, hair cortisol concentration; UDCA, ursodeoxycholic acid; C-section, Cesarean section; NR1, N-methyl-D-aspartic acid receptor 1; ERK1/2, extracellular signal-regulated kinase 1/2; PVN, paraventricular nucleus of the hypothalamus; PCF, prefrontal cortex; TLR-4, toll-like receptor 4; GLU, glutamate; FST, forced swimming test; PA, passive avoidance; NOR, novel object recognition; FO, fish oil; MS, maternal separation; MFGM, milk fat globule membrane; ELF, early limited formula; OPCML, opioid binding protein/cell adhesion molecule like.

#### 3.1.4. Antidepressants **

Prenatal serotonin/norepinephrine reuptake inhibitors (SSRIs) are antidepressants whose usage amplifies the risk of GI disorders in young individuals [53]. Pretreatment based on antibiotics in combination with fluoxetine alters the microbial profile. The joint effect was already ascertained in relative abundance of elemental beneficial entities and transcriptional responses [54,55].

#### 3.1.5. Exogenous Stressors ***

The widespread use of pesticides, organic compounds, and other pollutants has increased dramatically. It was shown that chlorpyrifos (CPF) [56], bisphenol A (BPA) [57], fluorescent nanosized polystyrene particles [58], polybrominated diphenyl ethers (PBDEs) [59], and triclosan (TCS) [60] are potent toxicant agents. The consequences are long-lasting even in small doses. Regardless of sex, these compounds have a neurotoxic profile dependent on the exposure period.

#### 3.1.6. Conventional and Alternative Methods ***

Neurotransmitters modulate the neuroendocrine network, as reflected by the peculiarities promoted following the food intake [61], whereas a lack of vitamins and minerals intensifies the GI permeability [62]. Cumulatively, this might further offer data on the underlying mechanism of gut dysbiosis [63]. Considering the constantly growing need to find a suitable adjuvant, some authors tested the potential of naturally derived compounds obtained from plants, even traditional Chinese medicine [64,65,66,67,68].

#### 3.2. Microflora and Behavioral Changing Factors *

Even though the microbiota is directly involved in the host’s eubiosis, it is also susceptible to a plethora of stressors. As anticipated, they have a pronounced influence on both newborn and mother’s homeostasis. This argument is strengthened by the studies conducted on infants (*n* = 2028) [69,70,71,72,73,74,75,76,77] and mothers (*n* = 255) [78,79]. Astsinki et al. [71] revealed using a clustering strategy that *Bacteroidetes* are associated with lower self-regulation capacity in contrast to a *Bifidobacterium*/*Enterobacteriaceae* cluster. They aimed at examining GI flora at 2.5 months of age and how it influences temperament at 6 months considering that *Bifidobacterium* and *Streptococcus* can shape positive emotionality. Loughman et al. [69] applied a different strategy by assessing the microbial associations on three occasions at 1, 6, and 12 months. *Prevotella* was significantly lower at 12 months compared with the other analyzed groups at 1 and 6 months. It is worth mentioning that this genus is practical as a predictor of antibiotic exposure. Similarly, Carlson et al. [73] analyzed GI microflora in full-term healthy individuals at 1 year and correlated them with Mullen Scale of Early Learning (MSEL) scores and brain anatomy (MRI) at 1 and 2 years of age. Of all the clusters, individuals populated by *Bacteroidetes* had more promising outcomes in terms of receptivity and expressive language than individuals reuniting *Faecalibacterium* and *Ruminococcaceae* groups. These performances were attributed to breastfeeding and vaginal delivery. There was a negative correlation between the α-diversity at 1 year and MSEL scores at 2 years, which was further probed in the study of Aatsinki but concentrated on opposing emotionality and fear reactivity. Xie et al. [70] confirmed the results of Carlson, revealing that *Faecalibacterium* is negatively related to emotional traits such as high-intensity pleasure and surgency but *Bifidobacterium* with perceptual sensitivity also at the age of 2. However, there was no association between gender and age linked to temperament and GI flora as reported through the Infant Behavior Questionnaire—Revised (IBQ-R) and Early Childhood Behavior Questionnaire (ECBQ). Kelsey et al. [74] found that abundance of *Bacteroides* and *Lachnospiraceae* are accountable for the connectivity of the neural network of the brain. The presence of *Slackia isoflavonivonvertens* could contribute to a longer gestational age, as recently demonstrated by Gough et al. [78], and *Rikenellaceae*, and *Dialister* abundance was linked with cortisol response to stress and negatively with *Bacteroides* [79]. Babies from families that have a low socioeconomic status (SES) displayed a higher α-diversity and, more precisely, a greater abundance of *Bifidobacterium* and *Megasphaera*. As highlighted above, an exogenous intake of probiotics reduces the risk of hospitalization due to *Staphylococcus aureus* [75]. Analyzing separate lines of research, maternal smoking has proven to be dangerous and interacts with cognitive functioning as measured with the Wechsler Preschool and Primary Scale of Intelligence (WPPSI-III). Presumably, this has to do with the relative abundance of *Enterobacter asburiae* [76]. The meconium microbiota is significantly influenced by anxiety, especially by a low level of *Enterococcaceae* [72], subsequently revealing that *Akkermansia*, *Lactococcus*, and *Oscillospira* ratio are positively associated with higher sociality scores [77].

#### 3.3. Fecal Microbiota Transplantation as a Preventive Measure *^,^**

Per normative disseminated by the World Health Organization (WHO), fecal microbiota transplantation (FMT) and microbial transfer therapy (MTT) are techniques dedicated to reconstructing the gut flora, implying that the transfer of the fecal samples from a healthy donor are instilled in a patient. FMT is usually applied to treat recurrent *Clostridium difficile* (rCDI) infections, while MTT focuses on metabolic deficiencies [80,81]. Presently, the overall efficiency of FMT in the setting is >90%, but it is dependent on the working protocols. The success rate is higher if multiple infusions with fecal quantities (>50 g) are scheduled. On the other hand, it strongly fluctuates based on the route of delivery, particularly after colonoscopy [82]. As another parameter that may be a contributor to rCDI treatment, it has been also successfully applied in a wide of other GI manifestations: irritable bowel syndrome (IBS), inflammatory bowel disease (IBD), metabolic and hepatic illnesses, graft and host disorders, disturbance of the GBA, and even multi-drug-resistant colonization [83,84].

Xu et al. [85] observed that constantly increasing concentrations of alcohol drove anxious and depressive behaviors in rodents. To consolidate the utility of this protocol, the authors subjected the models to prolonged FMT by oral gavage from healthy individuals. Sustaining the inflammasome hypothesis of depression, Zhang et al. [86] reported that the NLR family pyrin domain containing 3 (NLRP3) knock-out mice exhibits a significantly greater locomotor activity and increased *Lachnospiraceae*, *Ruminococcaceae*, and *Prevotellaceae* abundance to the detriment of *Bacteroides*. A three-day FMT reverse stress-induced dysbiosis alleviated both anxiety- and depression-like symptoms through the attenuation of the astrocytes-driven inflammation. Schmidt et al. [87] conducted an incomplete unilateral cervical spinal cord injury (SCI) in injured rats. Congruent with previous observations, FMT from healthy rats rescues anxious behavior and SCI-driven dysbacteriosis. *Faecalibacterium rodentium* exacerbates depressive-like symptoms in mice subjected to FTM [88] and in human patients as well [89]. Moreover, GI microbiota transfer from depressed humans induces a specific depressive-like phenotype in rats pretreated with antibiotics [90]. Tangent results were seen in germ-free mice (GFM) [91], which comprises the topic of a small sub-section later in this manuscript.

The role of FMT in restoring eubiosis in human patients has also been a subject of great interest. Mazzawi et al. [92] reported an improvement for 28 weeks in 13 IBS patients after single FMT through gastro-duodenoscopy in a small placebo-controlled study. Using Eysenck Personality Questionnaire-Neuroticism and Hospital Anxiety and Depression (HAD) tests, they observed only an improvement in HAD scores at three weeks post-intervention. Kurokawa et al. [93] conducted a small observational study seeking to examine 17 adult patients who underwent FMT via colonoscopy. They followed the psychiatric symptoms trend using Hamilton Rating Scale (HAM) for Anxiety (HAM-A) and Depression (HAM–D). Twelve patients had HAM-D scores at baseline ≥ 8 but in a subset of six patients normalized after one month. Five patients had HAM-A scores ≥14, but the scores of three patients decreased to normal. A Japanese open-label study of 10 IBS patients conducted by Mizuno et al. [94] revealed that donor material rich in *Bifidobacterium* stimulates the growth and expansion of strains. A good responder is one that achieves type 3 or 4 in the Bristol Stool Form Scale. Unfortunately, the amelioration of depressive symptoms lasted until the 12th week in comparison with the non-significant difference but slight progress in anxiety in week 4. In a similar manner, Lahtinen et al. [95] applied the Beck Depression Inventory (BDI), Beck Anxiety Inventory (BAI), IBS Quality of Life (IBS-QOL) questionnaires, and 15D in 49 patients enrolled for allogenic FMT via colonoscopy. Interestingly, the scores registered by patients were not statistically significant in total depression or anxiety. Likewise, Huang et al. [96] carried out a study by performing FMT from healthy donors in a group of 30 IBS refractory patients. They used IBS-QOL, HAM-D, and HAM-A to assess the evolution of non-gastrointestinal aspects at 1, 3, and 6 months after FMT. Noteworthy, FMT indeed contributed significantly to the IBS-QOL at 1 and 3 months but not after half a year. HAM-A and HAM-D scores indicate a significant amelioration only at 1 and 3 months. Among those 83 patients that participated in the study of Johnsen et al. [97] that aimed to evaluate the impact of FMT via the lower route in fatigue and quality of life of patients in IBS non-C, most of them experienced minimal improvements in IBS-QOL. The researchers used Fatigue Impact Scale (FIS) at 3, 6, and 12 months and IBS-QOL at 6 and 12 months, respectively. The sub-domains with the higher responsivity were interference with activity, body image, and relationships. Amelioration in the FIS score from baseline was seen at half a year in the FMT recipients post-intervention. However, the lack of presence of other functional disorders was used as predictor of profound prolonged response to 12 months. Another small case series brought to light the beneficial effect of FMT on depressed patients diagnosed with IBS via colonoscopy. It is still questionable whether decreased symptomatology was due to FMT or improvement of IBS [98].

#### 3.4. Microflora Involvement in Recurrent Implantation Failure

In contrast women who naturally achieve a pregnancy, those found in the pursuit of assisted reproductive technology (ART) are at higher risk of depression [99]. The fertility QOL of recurrent implantation failure (RIF) patients is dependent on numerous parameters [100], but the patterns are not related to the outcome of assisted pregnancy [101]. Patients experience symptoms of depression at one week post-intervention in the case of negative in vitro fertilization (IVF) [102]. Depression rates are high in women undergoing infertility treatment and even higher in those with IVF cycle failure [103]. As already documented, the vaginal tract is populated by the *Lactobacillus* genus, particularly with four strains, namely *Lactobacillus crispatus*, *Lactobacillus gasseri*, *Lactobacillus iners*, and *Lactobacillus jensenii*, which may counter the antagonistic activity of opportunistic bacteria. These microorganisms fulfill key roles and possess a eubiotic and immunological effect. These implications are centered on countering the proliferation of pathogens through SCFAs [104,105], antimicrobial peptides [106], and infections [107,108,109,110]. A possible dysbacteriosis causes consequences linked with different RIF stages [105,111], but there still are controversies regarding microbial translocations. It is known that endometrial fluid harbors *Lactobacillus* spp. and represents approximately 90% of all vaginal strains [112]. Subsequently, these data were contradicted and revealed the contrary [113]: *Lactobacillus* depletion causes repercussions reflected by poor clinical outcomes [112,113]. Two *Lactobacillus* species (*crispatus* and *iners*) are presently employed to categorize women with high chances of pregnancy [104]. 

#### 3.5. Antibiotics as a Vehicle of Cytokine Elevation and Gut Dysbacteriosis ***

Nyangahu et al. [114] suggested that perturbations of the maternal microbiome dictate neonatal adaptive immunity in pups. Vancomycin alters the α-diversity, leading to a pro-inflammatory cascade by triggering. Besides the elevation of IgG and IgM in breast milk, it also impacts lymphocyte numbers in pups, particularly CD4+ and follicular B cells. Ceftriaxone is another potent drug that kills most of the normal flora but, in combination with vancomycin, causes morphological villi changes, an increase in IgE, and a decrease of Ki67-/Muc2-positive cells as argued by Cheng et al. [115]. Not only did the *Proteobacteria* ascend as the dominant phylum, but the abundance of *Bacteroidetes*, *Firmicutes*, *Actinobacteria*, and *Deferribacteres* demonstrated a downward trend. This argument is supported by the recent data concerning ceftriaxone influences on α-diversity, as indicated twice by Cheng et al. [115,116]. *Bacteroidetes* almost entirely disappear from the feces following exposure to ceftriaxone as per Miao et al. [117], which is consistent with data disseminated by Cheng et al. [118]. Maternal antibiotic treatment (MAT) lowered the relative abundance of *Bacteroidetes* and *Firmicutes* in parallel with a proliferation of *Proteobacteria*. Consequently, it inflicted major phenotypic changes, notably significant intestinal injury and cytokine levels. More than that, MAT decreased the expression of vascular endothelial growth factor (VEGF), proliferating cell nuclear antigen positive-(PCNA), goblet cells, and tight junction proteins, as demonstrated by Chen et al. [118]. Additionally, Tormo-Badia et al. [119] noted a high balance of CD8+ T cells in the mesenteric lymph nodes (MLN) of pups from non-obese diabetic (NOD) females exposed to antibiotics during pregnancy in contrast to control pups. Continuing with this context, pups born to females that received oral antibiotics during gestation and postpartum displayed a high susceptibility risk towards vaccinia virus due to reduced interferon (IFN)-γ production by CD8^+^ T cells, according to Gonzalez-Perez and the co-authors [120]. The possible explanation for CD8^+^ T-cell disturbance might be the altered activation and expression of the T-cell receptor (TCR) that sustains cytokine production, as indicated by Gonzalez-Perez et al. [121]. Four markers identified by Benner et al. [122] seem to coordinate the robustness of the immune system after antibiotics administration: splenic T helper 17 cells and CD5^+^, CD4^+^ T cells in mesenteric lymph nodes as well as RORγT mRNA in the placenta.

#### 3.6. Germ-Free Nodels as a Novel Approach to Study AND and PDD ***

Among the first experimentations that aimed to offer a novel standpoint on how microorganisms inhabit the GI tract was the study of Sudo et al. [123]. The authors showed an elevated level of plasma CORT, an adrenocorticotropic hormone (ACTH), followed by a regressed expression in BNDF expression levels in GFM in both the cortex and hippocampus by comparison with the specific pathogen-free (SPF). *Bifidobacterium infantis* and depletion of *Escherichia coli* may correct HPA exaggerated stress response. Pessa-Morikawa et al. [124] outlined a range of gut microbial metabolites that cross the placenta into the fetal compartment to regulate the prenatal development process. *Escherichia coli* had an antagonistic role in maternal behavior maturation rather than directly impacting the infant. The presence of this pathogen interferes with IGF-1 signaling. Thus, Lee et al. [125] claimed that IGF-1 impairment resulted in malnourishment and consequential stunted growth. The conclusions of Sudo et al. [123] on the mechanism behind the transfer of feces were tested effective with “infant-type” *Bifidobacterium* strains. Luk et al. [126] generated a few years ago a simplified model community comprising human *Bifidobacterium* species displaying sex-dependent colonization reversible only in female GFM. The same authors again showed that a consortium of human-derived lactic acid bacteria strains belonging to *Bifidobacterium* (*longum subspecies infantis*, *bifidum*, *breve*, and *dentium*) are essential for microglial cell development. Otherwise, it might cause an elevation in synapse-promoting genes, markers of reactive microglia, and synaptic density in the hippocampus of GFM. Conclusively, *Bifidobacterium*-treated mice exhibit optimum synaptic density and neuronal activity, according to Luck et al. [127]. Martínez et al. [128] elegantly prioritize the effects of historical contingencies after they inoculated GFM in sequence with two contrasting microbial associations, with the mapping of communities six weeks later being related to colonization history. Koren [129], Ridaura [130], and Goodrich et al. [131] advanced how GFM inoculated with human feces from women in the first trimester resulted in several metabolic changes, such as weight gain and a slight reduction of insulin sensitivity [129], which are similar to the observations of inoculation from non-pregnant obese donors [130] but blocked by transplantation of *Christensenella* to prevent weight gain [131].

## 4. Conclusions

Due to the paucity of data, we were unable to conduct a quantitative meta-analysis of the studies that met the eligibility criteria. This is why we employed the best evidence to identify the key results and disentangle the possible interconnection between depression in all three stages and the disturbance of the gastrointestinal (GI)–vaginal microflora. Despite this impediment, we can argue that we have managed to provide an up-to-date comprehensive picture of the changes that might occur in the microbiota per searches and data identified following the centralization of the current evidence. Fortunately, this field of research benefits from an increasing interest reflected by the present mini-review, but future studies from an interdisciplinary approach are compulsory. We consider this manuscript to be the launching pad of the phase for deciphering the full potential behind the role of microbiota in depression and how this neuropsychiatric disorder can be modulated or triggered depending on the study design.

It can be concluded based on all aspects presented throughout this manuscript that depression during all three stages of pregnancy significantly influences the composition of GI microflora. Even though a possible dysbiosis may be corrected through numerous therapeutic approaches exclusively designed to date, there also exist hazardous factors that disrupt maternal–fetal interaction and implicitly the intrauterine life. In this context, the cognitive development of the infant is influenced, subjected to temperament and personality traits changes later in life. Thankfully, this field is evolving fast and has brought to life an alternative that proved its reliability in diminishing both anxiety- and depressive-like symptoms. However, depression is linked with associated with implantation failure, which could subsequently disturb microflora and diminish pregnancy chances. However, additional studies from our point of view are mandatory to establish links beyond theoretical stages. We can conclude that the studies contained in this mini-review are complementary, with the authors confirming/refuting the existing data. Although a transition to human patient experiences is being attempted, those performed on animal models proved to be groundbreaking. In order to create a defining parallel, the current field of research is evolving in tandem in terms of animal and human studies. A summary diagram can be found as seen below (Figure 1).

## Figures and Tables

**Figure 1 diagnostics-12-01576-f001:**
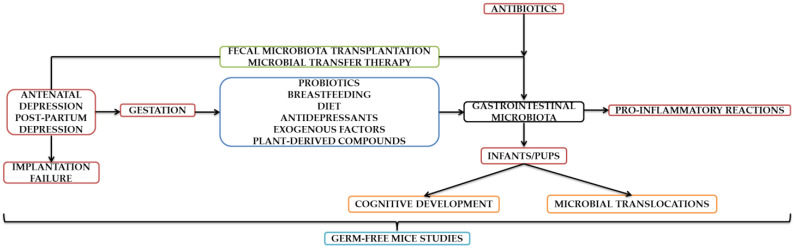
Schematic representation of the aspects contained in the Section 3.1, Section 3.2, Section 3.3, Section 3.4, Section 3.5 and Section 3.6.

**Table 1 diagnostics-12-01576-t001:** Stratification of studies in which are strictly reported changes that occurred in the gut flora.

Model	Hypervariable Region and Sequencer	Microbiota Changes	Reference
C57/Bl6 mice	V1–V3 16S rRNA MiSeq	*Bacteroides* ↓ *Parabacteroides* ↓	[14]
C57BL/6 mice	V3–V4 16S rDNA HiSeq 4000	*Bacteroides* ↑ *Alloprevotella* ↓*Muribaculaceae* ↓	[15]
Sprague-Dawley rats	V3–V4 16S rDNA NS	*Firmicutes* ↓ *Bacteroidetes* ↓	[16]
Women	V4 16S rRNA NS	*Faecalibacterium* ↓ *Phascolarctobacterium* ↓ *Butyricicoccus* ↓ *Lachnospiraceae* ↓ *Enterobacteriaceae* ↑	[17]
Women	V4 16S rRNA MiSeq	*Citrobacter* ↑ *Enterobacteriaceae* ↑*Weissella* ↑ *Lactobacillaceae* ↑ *Peptostreptococcaceae* ↓ *Veillonellaceae* ↓ *Gammaproteobacteria* ↓	[18]
Women	V4 16S rRNA MiSeq	*Proteobacteria* ↑ *Akkermansia* ↓ *Lactobacillus* ↓	[19]
Women	V4 16S rRNA MiSeq	*Lachnospiraceae* ↑ *Ruminococcaceae* ↑	[20]

↑, increase; ↓, decrease; NS, not specified.

## Data Availability

The datasets used and analyzed during the current study are available from the corresponding author on reasonable request.

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
