# Peer review of "An Updated Narrative Mini-Review on the Microbiota Changes in Antenatal and Post-Partum Depression"

_diagnostics, 2022, doi:10.3390/diagnostics12071576_

Round 1

Reviewer 1 Report

The manuscript by Doroftei et al is an exciting and timely manuscript addressing the microbiome issue in reproduction. Nevertheless, the manuscript is not easy to follow due to the lack of discrimination between animal models and humans. In practice, the animal models are listed in Table 1, while human models are outlined in Figure 1.  Reading in the text and particularly in materials and methods and other sections the reader is unable to find a clear cut between animals and humans and has to search the references for better comprehension of the text.
Therefore the manuscript needs revision with a clear description of the animal and human studies and also comprehensible divisions between the studies. In addition, a conclusion with differences and similarities should be written. 

Author Response

Dear Reviewer #1,

We would like to thank you very much for the positive feedback, interest, and time spent reviewing our manuscript. Per your instructions, we made the respective changes that can be found below:

Comments from the Reviewer:

The manuscript by Doroftei et al is an exciting and timely manuscript addressing the microbiome issue in reproduction. Nevertheless, the manuscript is not easy to follow due to the lack of discrimination between animal models and humans. In practice, the animal models are listed in Table 1, while human models are outlined in Figure 1.  Reading in the text and particularly in materials and methods and other sections the reader is unable to find a clear cut between animals and humans and has to search the references for better comprehension of the text.

Therefore the manuscript needs revision with a clear description of the animal and human studies and also comprehensible divisions between the studies. In addition, a conclusion with differences and similarities should be written.

Responses:

Dear Reviewer, we adopted a simplified strategy that would ease the reader’s effort to delineate the studies performed on human patients and experimental models. Specifically, we found it suitable to mark each subchapter from this mini-review with “*”, “**”, or “***” indicating either experiments conducted strictly on humans, from a combined perspective (humans + experimental models) or strictly on experimental. We summarize studies marked with “**” in a table that contains the key observations made by the authors (Table 2) based on the model (mice, rats, and humans) because as we reviewed the literature again and trying to respect your indications, we noticed a continuity of the logical thread from a combined perspective. Our tendency in the initial version sent was to separate each study specifically by the model (according to your instructions), but these studies are complementary. In a way, the evidence obtained on humans has been confirmed/refuted on animal models and vice versa. We have also added three new sentences in the Conclusions per your instructions. In summary, we made the following changes:

1) we revised once again the entire manuscript;

2) one new comprehensive table that contains the key observations from the sub-subchapter 3.1 based on the studied model;

3) one new subchapter regarding the involvement of the antibiotics in gut dysbiosis and subsequent pro-inflammatory reactions triggered;

4) one new subchapter regarding the use of germ-free mice as a novel approach to study antenatal and postpartum depression;

5) we updated Figure 1 which now contains the role of antibiotics and germ-free mice as well.

Kind regards and all the best,

Ovidiu-Dumitru Ilie

Reviewer 2 Report

I would like to thank the authors for submitting a nice mini-review on the topic of changes in the microbiome and their relevance to depression. 

For the most part it is a well written manuscript, but I would recommend it be read by a native English speaker as the grammar makes it quiet hard to understand. 

I would suggest some minor additions to the manuscript. 

The authors should discuss the role of antibiotics more, and given their high use in pregancy (pre and post) they should have more of an inclusion in the document. 

Also, more details on pregnancy in germ-free mice should be included if the data exhists, are germ-free mice more depressed post partum? Do they have more or less pups?. 

I think with these additions it is ready for publication, 

Author Response

Dear Reviewer #2,

We would like to thank you very much for the positive feedback, interest, and time spent reviewing our manuscript. Per your instructions, we made the respective changes that can be found below:

Comments from the Reviewer:

I would like to thank the authors for submitting a nice mini-review on the topic of changes in the microbiome and their relevance to depression.

For the most part it is a well written manuscript, but I would recommend it be read by a native English speaker as the grammar makes it quiet hard to understand.

Response:

Dear Reviewer, we revised once again the entire manuscript for typos and other errors. We also simplified each sentence by retaining the original meaning.

Comments from the Reviewer:

I would suggest some minor additions to the manuscript.

The authors should discuss the role of antibiotics more, and given their high use in pregancy (pre and post) they should have more of an inclusion in the document.

Response:

Per your instructions, we added a new subchapter titled:

3.5 Antibiotics as a vehicle of cytokine elevation and gut dysbacteriosis

“Nyangahu et al. [114] suggest that perturbations of the maternal microbiome dictate neonatal adaptive immunity in pups. Vancomycin alters the α-diversity, leading to a pro-inflammatory cascade by triggering. Besides the elevation of IgG and IgM in breast milk, it also impacts lymphocyte numbers in pups, particularly CD4+ and follicular B cells. Ceftriaxone is another potent drug that kills most of the normal flora but in combination with vancomycin cause morphological villi changes, an increase in IgE, and a decrease of Ki67-/Muc2-positive cells argued by Cheng et al. [115]. Not only did the Proteobacteria ascend as the dominant phylum, but the abundance of Bacteroidetes, Firmicutes, Actinobacteria, and Deferribacteres knew a downward trend. This argument is supported by the recent data concerning ceftriaxone influences on α-diversity as indicated twice by Cheng et al. [115,116]. Bacteroidetes almost entirely disappear from the feces following exposure to ceftriaxone per Miao et al. [117], which is consistent with data disseminated by Cheng et al. [118]. Maternal antibiotic treatment (MAT) lowered the relative abundance of Bacteroidetes and Firmicutes in parallel with a proliferation of Proteobacteria. Consequently, it inflicted major phenotypic changes, notably significant intestinal injury and cytokine levels. More than that, MAT decreased the expression of vascular endothelial growth factor (VEGF), proliferating cell nuclear antigen positive-(PCNA), goblet cells and of tight junction proteins as demonstrated by Chen et al. [118]. Additionally, Tormo-Badia et al. [119] note a high balance of CD8+ T cells in the mesenteric lymph nodes (MLN) of pups from non-obese diabetic (NOD) females exposed to antibiotics during pregnancy in contrast to control pups. Continuing with this context, pups born to females that received oral antibiotics during gestation and postpartum displayed a high susceptibility risk towards vaccinia virus due to reduced interferon (IFN)-γ production by CD8+ T cells according to Gonzalez-Perez and the co-authors [120]. The possible explanation for CD8+ T cell disturbance might be the altered activation and expression of T cell receptor (TCR) that sustain cytokine production as indicated by Gonzalez-Perez et al. [121]. Four markers identified by Benner et al. [122] seem to coordinate the robustness of the immune system after antibiotics administration; splenic T helper 17 cells and CD5+, CD4+ T cells in mesenteric lymph nodes, and RORγT mRNA in the placenta.”

Comments from the Reviewer:

Also, more details on pregnancy in germ-free mice should be included if the data exhists, are germ-free mice more depressed post partum? Do they have more or less pups?.

Response:

Per your instructions, we added a new subchapter titled:

3.6 Germ-free models as a novel approach to study AND and PDD

Among the first experimentations that aimed to offer a novel standpoint on how microorganisms inhabit the GI tract is the study of Sudo et al. [123]. The authors have shown an elevated level of plasma CORT, adrenocorticotropic hormone (ACTH), followed by a regressed expression in BNDF expression levels in GFM in both cortex and hippocampus by comparison with the specific pathogen-free (SPF). Bifidobacterium infantis and depletion of Escherichia coli may correct HPA exaggerated stress response. In this context, Pessa-Morikawa et al. [124] outline a range of gut microbial metabolites that cross the placenta into the fetal compartment to regulate the prenatal development process. Escherichia coli had an antagonistic role in maternal behavior maturation rather than directly impacting the infant. The presence of this pathogen interferes with IGF-1 signaling. Thus, Lee et al. [125] claimed that IGF-1 impairment resulted in malnourishment and consequential stunted growth. The conclusions of Sudo et al. [123] on the mechanism behind the transfer of feces were tested effective with “infant-type” Bifidobacterium strains. Luk et al. [126] generated a few years ago a simplified model community comprising human Bifidobacterium species displaying sex-dependent colonization reversible only in female GFM. The same authors again showed that a consortium of human-derived lactic acid bacteria strains belonging to Bifidobacterium (longum subspecies infantis, bifidum, breve, and dentium) are essential for microglial cell development. Otherwise, it might cause an elevation in synapse-promoting genes, markers of reactive microglia, and synaptic density in the hippocampus of GFM. Conclusively, Bifidobacterium-treated mice exhibit optimum synaptic density and neuronal activity according to Luck et al. [127]. Martínez et al. [128] elegantly prioritize the effects of historical contingencies after they inoculated GFM in sequence with two contrasting microbial associations, the mapping of communities six weeks later being related to colonization history. Koren [129], Ridaura [130], Goodrich et al. [131] advanced how GFM inoculated with human feces from women in the first trimester resulted in several metabolic changes, such as weight gain and a slight reduction of insulin sensitivity [129], similar to the observations when applied the inoculation from non-pregnant obese donors [130], but blocked by transplantation of Christensenella to prevent weight gain [131].

Comments from the Reviewer:

I think with these additions it is ready for publication.

Response:

Dear Reviewer, it was a great honor for us, and thank you.

Kind regards and all the best,

Ovidiu-Dumitru Ilie

Round 2

Reviewer 1 Report

I review the revised version and I accept the present form.

Author Response

Dear Reviewer #1,

We would like to thank you very much for the positive feedback, interest, and time spent reviewing our manuscript. It was an actual honor for us.

Kind regards and all the best,

Ovidiu-Dumitru Ilie